# Association between enrolment with a Primary Health Care provider and amenable mortality: A national population-based analysis in Aotearoa New Zealand

**Pushkar Silwal** [1,2]*, **Maite Irurzun Lopez**[1], **Megan Pledger**[1], **Jacqueline Cumming**[1], **Mona Jeffreys**[1]

1 Te Hikuwai Rangahau Hauora | Health Services Research Centre, Te Herenga Waka-Victoria University of Wellington, Wellington, New Zealand, 2 School of Population Health, Faculty of Medical and Health Sciences, University of Auckland, Auckland, New Zealand

* p.silwal@auckland.ac.nz

## Abstract

**Data Availability Statement:** All relevant aggregated data are within the manuscript and its Supporting Information files. The data was

### Introduction

In Aotearoa New Zealand, being enrolled with a Primary Health Care (PHC) provider furnishes opportunities for lower cost access to PHC, preventative care and secondary health care services, and provides better continuity of care. We examine the characteristics of populations not enrolled, and whether enrolment is associated with amenable mortality.

### Method

We retrieved records of all deaths registered 2008 to 2017 in Aotearoa New Zealand, which included demographic and primary cause of death information. Deaths were classified as premature (aged under 75 years) or not, and amenable to health care intervention or not. The Primary Health Organisation (PHO) Enrolment Collection dataset provided the PHC enrolment status. Logistic regression was used to estimate the risk of amenable deaths by PHO enrolment status, adjusted for the effects of age, sex, ethnicity and deprivation.

### Results

A total of 308,628 mortality records were available. Of these, 38.2% were premature deaths, and among them 47.8% were amenable deaths. Cardiovascular diseases made up almost half of the amenable deaths. Males, youths aged 15–24 years, Pacific peoples, Māori and those living in the most socio-economically deprived areas demonstrated a higher risk of amenable mortality compared to their respective reference category. One in twenty (4.3%) people who had died had no active enrolment status in any of the calendar years in the study. The adjusted odds of amenable mortality among those not enrolled in a PHO was 39% higher than those enrolled [Odds Ratio = 1.39, 95% Confidence Interval 1.30–1.47].

obtained from the Ministry of Health, National
Collections Division, after following the standard
data request process and ethics approval. The
details of the dataset (e.g., name, variables) are
provided in the methodology section. The dataset
can be requested from the MOH through this link -
https://www.health.govt.nz/nz-health-statistics/
access-and-use/data-request-form.

**Funding:** This study was conducted as part of the
five-year Health Research Council NZ funded
primary health care programme (HRC 18/667)
"Enhancing primary health care services to
improve health in Aotearoa New Zealand". The
funders had no role in study design, data collection
and analysis, decision to publish, or preparation of
the manuscript.

**Competing interests:** The authors have declared
that no competing interests exist.

## Implications

Being enrolled in a PHC system is associated with a lower level of amenable mortality.
Given demonstrated inequities in enrolment levels across age and ethnic groups, efforts to
improve this could have significant benefits on health equity.

## Introduction

Primary Health Care (PHC) is fundamental to many health care systems worldwide, and integral to universal health coverage and the Sustainable Development Goals [1]. In general, it has
been demonstrated that investments in PHC improve access, equity, outcomes, performance
and accountability of health systems [1–3].

In Aotearoa New Zealand, the Primary Health Care Strategy 2001 identified a strong PHC
system as a guiding approach to improving health and reducing health inequities; that Strategy
has set the direction and vision for PHC services for the past two decades [4]. In contrast to
the publicly funded hospital services [5], PHC is provided through a range of private providers,
including general practitioners (GPs), practice nurses and pharmacists. Primary Health Organisations (PHOs), with funding channelled through District Health Boards (DHBs), are the
institutions responsible for ensuring the delivery of PHC services [4]. GP services are partly
subsidised from government capitation funding, and partly through out-of-pocket payments,
which are lower for people who are enrolled with a GP practice [4]. In addition to accessing
lower cost care, being enrolled confers benefits such as improved coordination of care within
the PHC system, and between PHC and secondary health care services.

From a patient's perspective, individuals enrol with the practice as that is where they fill in
the paperwork and accrue the benefits; from the funder's point of view, patients enrol with a
PHO. The capitation funding to PHOs and practices is proportional to the number of people
enrolled in the practice. PHOs disburse the funding given to them by the DHBs, and oversee
their practices. Previously, we have identified that approximately 6% of the eligible population
in Aotearoa New Zealand were not enrolled with a PHO, that the proportion of the population
not enrolled is growing, and that inequities across groups were not being addressed [6]. Low
enrolments were particularly evident for Māori, the indigenous population of Aotearoa New
Zealand; for young adults, and for people living in the Auckland region.

There is limited empirical evidence that shows the extent to which enrolment with a PHC
provider is related to good health outcomes. This could occur through one or more pathways,
including access to screening and other preventative health measures, management of long-term conditions, access to referral services and more affordable health care. Another core component of the enrolment system is continuity of care. Systematic reviews have found that continuity of care is associated with lower risks of ambulatory care sensitive hospitalisations [7]
and lower all-cause mortality [8], although not all studies consistently report the latter [9]. We
have found no literature that has directly assessed the relationship of being registered or
enrolled with a GP and subsequent mortality risk. The purpose of this analysis is to quantify
the association of not being enrolled with a PHO in Aotearoa New Zealand on amenable mortality, i.e. mortality for conditions that are amenable to health care interventions [10, 11].

## Materials and methods

In the absence of a national PHC database, we conducted an observational study using
linked datasets of PHO enrolments and mortality from 2008 to 2017. We estimated the

unadjusted and adjusted risk of amenable death by enrolment status through logistic regression analyses.

## Data sources

We obtained linkable data from the National Collections Division, Ministry of Health (MoH), New Zealand in 2020. These were:

a. **The PHO Enrolment Collection dataset**, which is a national collection of PHC System patient enrolment data in Aotearoa New Zealand established in 2005 [12]. The dataset is incomplete for the earlierst years of collection; therefore, we restricted the study to data from 2008 and limited it to before 2018 to match the Mortality Collection data set. The PHO enrolment data set only contains info about people 1) who have been enrolled but are not currently enrolled or 2) are currently enrolled. It does not contain info about people who have never been enrolled. The data was output for each quarter of each year.

b. **The Mortality Collection dataset**, which provides a central collection of all deaths registered in Aotearoa New Zealand with their primary cause of death (COD) and basic socio-demographic characteristics [11]. The COD information comes from multiple sources, including police reports, hospital discharge summaries and death certificates. This can lead to delays in finalising the dataset. Therefore, the latest complete year of data we could obtain at the time of analysis was 2017.

Ethical approval for this study was obtained from the Human Ethics Committee, Victoria University of Wellington (Ref: #29343).

## Data linkage and processing

Each dataset contains two identifiers, a national health identifier (NHI) and a master national health identifier (mNHI). Although each person should only have one NHI, this is not always the case, e.g. a person seeking care, does not know their NHI, and there has been an error in the recording of their name or date of birth. In that instance, a second or further NHI may be allocated, if their previous record cannot be identified in the NHI database. When duplicates are resolved, their unique identifier is recorded in the mNHI. For the purposes of research, the MoH supplied datasets with the NHI and mNHI encrypted, and are referred to here as the eNHI and emNHI.

The Mortality dataset provided the base population. Out of the total 309,777 death records available from 2008 to 2017, we excluded one case with a duplicated record and 1,148 cases with the date of death (DOD) recorded prior to 2008, resulting in 308,628 mortality cases. We linked the mortality and enrolment datasets using emNHI to obtain the enrolment status information of the mortality cases. The linked dataset generated an additional 1,921 records/cases in the Mortality dataset. This was because multiple eNHIs were associated with the same emNHI. We retained the records having a common eNHI in both datasets, resulting in a study population of 308,628.

## Enrolment status

PHO enrolment status is the primary predictor of interest in this analysis. Individuals can choose to enrol at any PHC provider, and can only be enrolled at one provider at a time. They remain enrolled as long as they have had a consultation with that provider within the previous three years. If they have not had a consultation with that provider in the last three years, their enrolment would lapse, unless they reaffirm their wish to remain enrolled, and the three year

clock starts again. People can choose not to enrol with a PHC provider and they are still entitled to receive services, but they will not benefit from additional or subsidised services.

Based on the PHO record of the individual cases, an overall dichotomous enrolment status variable was generated. People coded as enrolled had at least one active enrolment status during the study period; those coded as not enrolled had no active enrolment status or no record in the PHO dataset. The literature reports that the use of PHC services changes towards the end of life [13]. As a sensitivity analysis, we explored differences in the pattern of PHO enrolment and mortality outcomes when the enrolment status in the year of death is specified. The dichotomous variable included those enrolled at death, defined as having active enrolment status in the year of death and those not enrolled at death were those with no active enrolment in the year of death (irrespective of the active status in other years), or having no record in the PHO dataset.

### Socio-demographic variables

Age, sex, prioritised ethnicity and area-level deprivation were considered as potential confounder variables. The first three are sourced from the Mortality Collection dataset and the measure of deprivation is from an Index of Multiple Deprivation dataset. Age at death was grouped into eight categories as 'Under 1', '01–04', '05–14', '15–24', '25–44', '45–64', '65–74', and '75+'. 'Under 1' was considered a separate category to address the potential differential enrolment practices for this age group. The broad category of '75+' represents the categories of death not considered premature [14]. Sex was classified as male, female and unknown. For the purpose of modelling, we recoded "unknown" (n = 66) to male (n = 154,883) which was the largest sex group. In Aotearoa New Zealand, analysis of health data by ethnicity commonly relies on a measure known as prioritised ethnicity [15]. A person is allocated to only one high-level ethnic grouping in the following order of priority: Māori, Pacific Peoples, Asian and European/Other. Due to small sample sizes of the Asian category in our study, particularly for the unenrolled cases, the latter two categories were combined into a Non-Māori, Non-Pacific (NMNP) group. Deprivation status was assigned using the Index of Multiple Deprivation (IMD) [16], based on each individual's place of residence at the time of death as recorded in the Mortality Collection dataset. The index uses 28 indicators from the New Zealand census that relate to income, employment, crime, housing, health, education, and access to public services to generate a score for a geographic area (containing, on average, 761 people) [16]. These scores generate quintiles which are assigned to each person in the respective geographic area, with quintile 1 indicating the least deprived, and quintile 5 the most deprived, area. People whose place of residence was missing had their deprivation level put into a separate category labelled 'Missing'.

### Outcome variables

The primary outcome variable was any death registered between 2008 and 2017. We categorised deaths in two groups: premature deaths (aged under 75 years at the date of death) and those aged 75 and above. Premature deaths were further classified as Amenable Mortality (AM) or non-Amenable Mortality (nAM), following the MoH definitions [14]. Amenable mortality refers to premature deaths that could potentially be avoided given effective and timely health care, i.e. early deaths from causes (diseases or injuries) for which effective health interventions exist [14]. The MoH has a defined code list for AM, based on the primary cause of death, see S1 Appendix. The MoH AM code lists are updated every five years, the latest being the 2016 version [14]. As per the guideline [17], we used the 2012 version for 2008 to 2009 data and the 2016 version for the 2010–2017 dataset. For the purposes of analysis,

amendable deaths were further classified into six categories–Cancer, Cardiovascular Diseases, Infection, Injuries, Maternal and Infant, and Others [11].

### Statistical analysis

We conducted descriptive statistical analysis for the study population. Counts and percentages were used to summarize the mortality characteristics, enrolment status and background characteristics. Logistic regression was used to look at the association between amenable and non-amenable mortality and each of the socio-demographic variables as well as the enrolment variable (bivariate analysis). This was followed by logistic regression with all the variables included (multivariate analysis).

The R software's 'glm' with binomial link logit function (RStudio Version 1.2.5019) was employed for the analysis [18]. The R package "car" (version 3.0–7) was used to test for multicollinearity [19], and the decision criteria were based on the Variance Inflation Factor (VIF) with the recommended VIF cut-off of $< 3$ for main effects [20]. We report minimal correlation between predictors in the model (VIF<1.10) for all the model terms except the enrolment status variable for which it was moderate (VIF = 1.53). All tests were two-sided with a 5% significance level.

We conducted a sensitivity analysis with the variable 'enrolment status at death' in the logistic regression model, keeping all other predictor variables and the model structure the same.

## Results

### Mortality characteristics

The mortality dataset for the years 2008 to 2017 provided a total of 308,628 unique mortality records. Of these, 190,637 (61.8%) were those aged 75 and above, and 117,991 (38.2%) were premature deaths. Amenable deaths (56,414) made up 18.3% of all registered deaths and 47.8% of premature deaths. Among those died from amenable causes, one in ten (8.9%) were not enrolled with a PHO. Similarly, 60% of them were male, almost 80% were aged 45–74 years, 70% were Non-Māori, Non-Pacific and almost 30% each lived in deprivation quintiles 3 and 4 (Table 1).

The details of the amenable deaths in terms of the major primary cause categories are provided in S2 Appendix. The cardiovascular diseases category represents almost half (45.3%) of the total amenable deaths followed by cancers (23.3%), injuries (15.0%), maternal and infant disorders (5.2%) and infections (0.8%). Māori represents 22.0% in the total amenable deaths. In comparison to other ethnicity groups, they died more from Injuries (16.2% versus 9.1% among Pacific peoples and 15.3% NMNP) and least from Cancers (14.5% versus 18.2% among Pacific peoples and 26.6% among NMNP). Similarly, the causes of amenable deaths also vary across the socio-economic deprivation groups, with those living in the least deprived areas representing highest proportion of Cancers deaths (33.1%) and lowest proportion of the cardiovascular deaths (36.7%).

### Characteristic of the not enrolled population in study

Overall, 4.3% were not enrolled for at least one calendar year during the nine years of the study period which is 9.2% among the premature deaths. Similarly, 5.8% of the overall and 10.8% of the premature mortality cases were not enrolled in the year of death. (S3 Appendix summarises characteristics of study population (premature deaths) according to enrolment in the year of death).

**Table 1. Socio-demographic characteristics of the study population by premature and amenable death status.**

| | Premature deaths[1] (0–74 years) | | | | 75 years and above deaths | | Total | |
| | Non-amenable[1] | | Amenable[1] | | | | | |
| | N | Col % | N | Col % | N | Col % | N | Col % |
|---|---|---|---|---|---|---|---|---|
| **Enrolment status (overall)** | | | | | | | | |
| Enrolled | 55,729 | 90.5 | 51,368 | 91.1 | 188,311 | 98.8 | 295,408 | 95.7 |
| Not enrolled | 5,848 | 9.5 | 5,046 | 8.9 | 2,326 | 1.2 | 13,220 | 4.3 |
| **Sex** | | | | | | | | |
| Female | 26,301 | 42.7 | 22,680 | 40.2 | 104,748 | 54.9 | 153,729 | 49.8 |
| Male | 35,224 | 57.2 | 33,720 | 59.8 | 85,889 | 45.1 | 154,833 | 50.2 |
| Unidentified | 52 | 0.1 | 14 | 0.0 | 0 | 0.0 | 66 | 0.0 |
| **Age (years)** | | | | | | | | |
| Under 1 | 4,462 | 7.2 | 2,844 | 5.0 | - | - | 7,306 | 2.4 |
| 01–04 | 464 | 0.8 | 125 | 0.2 | - | - | 589 | 0.2 |
| 05–14 | 439 | 0.7 | 249 | 0.4 | - | - | 688 | 0.2 |
| 15–24 | 1,299 | 2.1 | 2,231 | 4.0 | - | - | 3,530 | 1.1 |
| 25–44 | 4,836 | 7.9 | 5,458 | 9.7 | - | - | 10,294 | 3.3 |
| 45–64 | 23,417 | 38.0 | 21,534 | 38.2 | - | - | 44,951 | 14.6 |
| 65–74 | 26,660 | 43.3 | 23,973 | 42.5 | - | - | 50,633 | 16.4 |
| 75+ | - | - | - | - | 190,637 | 100.0 | 190,637 | 61.8 |
| **Prioritised Ethnicity** | | | | | | | | |
| Māori | 11,750 | 19.1 | 12,432 | 22.0 | 8,797 | 4.6 | 32,979 | 10.7 |
| Pacific people | 4,041 | 6.6 | 4,674 | 8.3 | 4,117 | 2.2 | 12,832 | 4.2 |
| NMNP[2] | 45,786 | 74.4 | 39,308 | 69.7 | 177,723 | 93.2 | 262,817 | 85.2 |
| **Area level deprivation (quintiles)** | | | | | | | | |
| Least deprived | 6,095 | 9.9 | 5,078 | 9.0 | 19,531 | 10.2 | 30,704 | 9.9 |
| 2 | 13,328 | 21.6 | 11,349 | 20.1 | 51,784 | 27.2 | 76,461 | 24.8 |
| 3 | 18,224 | 29.6 | 16,437 | 29.1 | 62,437 | 32.8 | 97,098 | 31.5 |
| 4 | 16,392 | 26.6 | 15,935 | 28.2 | 43,637 | 22.9 | 75,964 | 24.6 |
| Most deprived | 4,753 | 7.7 | 5,049 | 8.9 | 6,819 | 3.6 | 16,621 | 5.4 |
| Missing | 2,785 | 4.5 | 2,566 | 4.5 | 6,429 | 3.4 | 11,780 | 3.8 |
| **Total** | **61,577** | **100.0** | **56,414** | **100.0** | **190,637** | **100.0** | **308,628** | **100.0** |

Notes: 1) The definitions of premature, amenable, and non-amenable deaths are as per the MOH definitions. 2) NMNP is the Non-Māori, Non-Pacific population

The enrolled and not enrolled are near equally likely to be male although males dominate in both groups (approx. 59%). The not-enrolled are more likely to be under 1 (62% vs .5%) and less likely to be over 45 (26% vs 86%) compared to the enrolled. In comparison, the not enrolled are more likely to be Māori and Pacific peoples but the Non-Māori, Non-Pacific peoples still make up the majority of the deaths (61%). The not enrolled are more likely to be in the most deprived category (11% vs 8%) while the enrolled are more likely to be in the lesser deprived categories.

Within the group of those not enrolled at some point (n = 10,894), a majority (98.8%) had never been enrolled in the study period. The bivariate analysis shows that a higher proportion of those not enrolled were males (58.9%), aged under 1 year (61.9%) followed by those aged 45–64 years (16.2%), Non-Māori Non-Pacific (61.0%) and those living in the deprivation quintiles 3 and 4 (25.0% each) (Table 2).

**Table 2. Socio-demographic characteristics of the study population (premature deaths) by enrolment status.**

| | Enrolled | | Not-enrolled | | Chi squared* (p value) |
|---|---|---|---|---|---|
| | N | Col % | N | Col % | |
| **Sex** | | | | | |
| Female | 44506 | 41.6 | 4475 | 41.1 | 0.33 |
| Male | 62591 | 58.4 | 6419 | 58.9 | |
| **Age (years)** | | | | | |
| Under 1 | 560 | 0.5 | 6,746 | 61.9 | |
| 01–04 | 563 | 0.5 | 26 | 0.2 | 0.001 |
| 05–14 | 663 | 0.6 | 25 | 0.2 | |
| 15–24 | 3,150 | 2.9 | 380 | 3.5 | |
| 25–44 | 9,505 | 8.9 | 789 | 7.2 | |
| 45–64 | 43,191 | 40.3 | 1,760 | 16.2 | |
| 65–74 | 49,465 | 46.2 | 1,168 | 10.7 | |
| **Ethnicity** | | | | | |
| Māori | 21,487 | 20.1 | 2,695 | 24.7 | 0.001 |
| Pacific people | 7,156 | 6.7 | 1,559 | 14.3 | |
| Non-Māori Non-Pacific | 78,454 | 73.3 | 6,640 | 61.0 | |
| **Area level deprivation (quintiles)** | | | | | |
| Least deprived | 10,304 | 9.6 | 869 | 8.0 | 0.001 |
| 2 | 22,892 | 21.4 | 1,785 | 16.4 | |
| 3 | 31,930 | 29.8 | 2,731 | 25.1 | |
| 4 | 29,553 | 27.6 | 2,774 | 25.5 | |
| Most deprived | 8,616 | 8.0 | 1,186 | 10.9 | |
| Missing | 3,802 | 3.6 | 1,549 | 14.2 | |
| **Total** | **107,097** | **100.0** | **10,894** | **100.0** | |

**Note:** Enrolled includes those having at least one active enrolment status during the study years; Not-enrolled includes those having no PHO record or having no active enrolment status during the study years.

* Chi squared test of association between the enrolment status and the population sub-groups (characteristics).

## Relationship between primary health care enrolment and amenable mortality

The adjusted odds of amenable mortality among those not enrolled are 38.5% higher [OR = 1.38, 95% CI 1.30–1.47] than those enrolled (Table 3). Similarly, males [OR 1.1, 95% CI 1.07–1.12] compared to females, youths aged 15–24 years [OR 6.37, 95% CI 5.18–7.88] compared to those aged 01–04 years, Pacific People and Māori [OR 1.32, 95% CI 1.25–1.38 and 1.22, 1.19–1.26 respectively] compared to non-Māori, non-Pacific people, and those living in the most deprived areas [OR = 1.12, 95% CI 1.06–1.19] compared to those in the least deprived areas are more likely to die from amenable causes.

The OR for the enrolment variable changed from 0.94 in the bivariate model to 1.38 in the multi-variate model. To explore which variable caused this change in size of the OR, a model with mortality and enrolment status was fitted with each of the socio-demographic variables in turn. This showed that age group was the variable that affected this change moving the OR from 0.94 to 1.34 while for the other variables the change was less than 0.1.

Table 4 shows estimates of the association between not being enrolled and amenable mortality, stratified by age group, with and without adjusting for other socio-demographic

**Table 3. Socio-demographic characteristics of amenable mortality cases and ORs of amenable deaths versus non-amenable deaths.**

| | Amenable deaths | | Unadjusted (bivariate) estimates | | | | Adjusted (multi-variate) estimates | | | |
|---|---|---|---|---|---|---|---|---|---|---|
| | N | %[1] | OR | 95% CI | | P-value | OR | 95% CI | | P-value |
| **Total** | 56,414 | 47.8 | | | | | | | | |
| **Enrolment status (overall)** | | | | | | | | | | |
| Enrolled | 51,368 | 48.0 | Ref | | | | Ref | | | |
| Not enrolled | 5,046 | 46.3 | 0.94 | 0.90 | 0.97 | <0.001 | 1.38 | 1.30 | 1.47 | <0.001 |
| **Sex** | | | | | | | | | | |
| Female | 22,680 | 46.3 | Ref | | | | Ref | | | |
| Male[2] | 33,720 | 48.9 | 1.11 | 1.08 | 1.13 | <0.001 | 1.10 | 1.07 | 1.12 | <0.001 |
| **Age (years)** | | | | | | | | | | |
| Under 1 | 2,844 | 38.9 | 2.37 | 1.94 | 2.91 | <0.001 | 1.80 | 1.46 | 2.23 | <0.001 |
| 01–04 | 125 | 47.3 | Ref | | | | Ref | | | |
| 05–14 | 249 | 36.2 | 2.11 | 1.64 | 2.71 | <0.001 | 2.12 | 1.65 | 2.73 | <0.001 |
| 15–24 | 2,231 | 63.2 | 6.38 | 5.19 | 7.89 | <0.001 | 6.37 | 5.18 | 7.88 | <0.001 |
| 25–44 | 5,458 | 53.0 | 4.19 | 3.44 | 5.14 | <0.001 | 4.28 | 3.51 | 5.26 | <0.001 |
| 45–64 | 21,534 | 47.9 | 3.41 | 2.81 | 4.18 | <0.001 | 3.60 | 2.96 | 4.41 | <0.001 |
| 65–74 | 23,973 | 42.5 | 3.34 | 2.75 | 4.09 | <0.001 | 3.62 | 2.98 | 4.44 | <0.001 |
| **Prioritised Ethnicity** | | | | | | | | | | |
| Māori | 12,432 | 51.4 | 1.23 | 1.20 | 1.27 | <0.001 | 1.22 | 1.19 | 1.26 | <0.001 |
| Pacific people | 4,674 | 53.6 | 1.35 | 1.29 | 1.41 | <0.001 | 1.31 | 1.25 | 1.38 | <0.001 |
| NMNP[3] | 39,308 | 46.2 | Ref | | | | Ref | | | |
| **Area level deprivation (quintiles)** | | | | | | | | | | |
| 1 Least deprived | 5,078 | 45.4 | Ref | | | | Ref | | | |
| 2 | 11,349 | 46.0 | 1.02 | 0.98 | 1.07 | 0.34 | 1.01 | 0.96 | 1.06 | 0.74 |
| 3 | 16,437 | 47.4 | 1.08 | 1.04 | 1.13 | <0.001 | 1.05 | 1.00 | 1.09 | 0.04 |
| 4 | 15,935 | 49.3 | 1.17 | 1.12 | 1.22 | <0.001 | 1.09 | 1.04 | 1.14 | <0.001 |
| 5 Most deprived | 5,049 | 51.5 | 1.28 | 1.21 | 1.35 | <0.001 | 1.12 | 1.06 | 1.19 | <0.001 |
| Missing | 2,566 | 48.0 | 1.11 | 1.04 | 1.18 | <0.001 | 1.00 | 0.93 | 1.07 | 0.95 |

**Notes:** 1) The percent is the proportion of total premature deaths (n = 117991). 2) Unidentified sex has been added to the male category for analysis. 3) NMNP is the Non-Māori, Non-Pacific population.

**Table 4. The association between enrolment and amenable mortality, stratified by age.**

| Age group | Estimates for enrolment only model[1] | | Estimates for enrolment—full model[1,2] | |
|---|---|---|---|---|
| | Odds ratio | P-value | Odds ratio | P-value |
| Under 1 | 4.97 | <0.01 | 4.90 | <0.01 |
| 01–04 | 0.88 | 0.80 | 0.61 | 0.35 |
| 05–14 | 0.43 | 0.10 | 0.44 | 0.12 |
| 15–24 | 0.62 | <0.01 | 0.70 | <0.01 |
| 25–44 | 0.90 | 0.17 | 0.95 | 0.49 |
| 45–64 | 1.47 | <0.01 | 1.43 | <0.01 |
| 65–74 | 1.43 | <0.01 | 1.39 | <0.01 |

**Note:** 1) Enrolment variable reference category–Enrolled. 2) Adjusted variables in the full model: sex, ethnicity and deprivation. Each age group modelled separately

variables. Where the odds ratios were significant, they were highest for those aged Under 1 year (OR = 4.90, p<0.01) and lowest among 15–24 years group (OR = 0.70, p<0.01).

The sensitivity analysis, where enrolment status in the year of death is the primary predictor, supports the finding that the adjusted odds of amenable mortality among those not enrolled is higher than those enrolled. The corresponding adjusted estimate for enrolment at death [OR = 1.35, 95% CI 1.28–1.42] is 35% higher among those not enrolled in the year of death compared to those who had an active enrolment status at death.

## Discussion

Preventing avoidable ill health, and ultimately avoiding amenable mortality, is a goal of any health system, and a marker of health system performance. Our study has found that after adjusting for socio-demographic variables, people who died from amenable causes (preventable through health care interventions) were more likely to have been out of the PHO network, and not enrolled in the year of death. This analysis is in line with our initial hypothesis that not being enrolled and, thus, missing out on the benefits that enrolling provides, is associated with negative health outcomes. To our knowledge, this is the first empirical evidence to show the relationship between PHC enrolment and mortality outcomes in Aotearoa New Zealand. Furthermore, we have documented inequities in enrolment by age, ethnicity and area-level deprivation. In particular, the proportion of deaths that are occurring in people who are unenrolled is particularly high for Pacific peoples, with the odds of amenable mortality being over 40% higher than among non-Māori/non-Pacific people. Similar patterns, although to a lesser extent, are seen for Māori.

Our findings suggest there is opportunity to reduce mortality through a strengthening of the PHC system, but alternative explanations of our findings must first be considered carefully. There are various possible reasons why people who are not enrolled may have higher rates of amenable mortality. Firstly, we acknowledge that there may be residual confounding in our study. One source of this could be based on need. Given that capitation payments are based only on age and sex [21], and are not weighted by any measure of health need, these payments do not adequately fund practices with high levels of higher needs people [22]. Providers are not required to enrol new patients; this can be applied to all new applicants, if a provider does not have capacity to take on new patients (known as "closed books"). This may alo occur on an individual basis, where a provider chooses to take on only some patients, with practices "cherry-picking" which patients to enrol, i.e. refusing to enrol people with chronic health condisitons.

It is also important to acknowledge the demonstrated institutional racism within the health care system in Aotearoa New Zealand [23]. Experience of interpersonal racism has been demonstrated as an independent risk factor for chronic illness [24], it is also likely to reduce a person's likelihood of wanting to enrol with a provider. This could explain in part the differential results that we see by ethnicity, where Māori are less likely to enrol with a PHC provider, and experience higher rates of amenable mortality. The health system in Aotearoa New Zealand is currently undergoing major reforms, and a Māori Health Authority is being established. Although this is unlikely to have significant impact on amenable mortality in the short term, the hope is that the system can be reoriented to better meet the needs of Māori.

In addition to residual confounding, there are other limitations of this work. We used enrolment with a PHO as a marker of access to PHC. As noted above, patients can access services as "walk-ins", without being enrolled; but we have not been able to quantify this.

Finally, there is a possibility that there are differential errors in matching across the two datasets that we used. The status of being enrolled or not being enrolled depends on being able

to link a respondent in the two datasets together, or not, via the emNHI. However, a person may have acquired two (or more) eNHIs which may not have been resolved to identify one person with a single emNHI. If this happens, so that there is one emNHI in the mortality dataset and a different one in the enrolment dataset then we would not be able to link the data correctly to determine that person's correct enrolment status. This may happen differentially by ethnic status where some surnames maybe more complex or unfamiliar. We investigated this using the enrolment dataset where we calculated for each ethnicity, the number of eNHIs per emNHI. For non-Māori, non-Pacific people, 0.9% had more than one eNHI per emNHI, for Māori it was 1.5% and for Pacific Peoples it was 2.3%. This suggests that some of the difference in enrolment status between cultures is due to differences in resolving multiple eNHIs to a single emNHI.

There are few studies with which to compare our work. A study from Canada found that immigrants (as opposed to long term residents of Canada) were less likely to be enrolled in primary care practices, with the consequence that they are less likely to benefit from the (presumed) higher quality care offered by integrated services [25]. Although we have not investigated immigrants, we have demonstrated inequities in enrolment by ethncity, which was particularly notable for Māori, as well as for Pacific peoples. A further Canadian study showed that patients with diabetes enrolled with a Primary Care Network had a lower rate of hospital admissions or emergency department visits for diabetes-specific ambulatory care sensitive conditions [26]. Although not fully comparable with PHO enrolment in Aotearoa New Zealand, some of the features in the Canadian study are similar to those that enrolment with a PHO would confer, including additional funding to the practice (which would occur through capitation funding) and integration across the PHC team, which is more likely to be accessible to those enrolled in a PHO.

We found that 4% of those recorded as deceased were not enrolled with a PHO in the study period. This is lower than the 6% of the population not being enrolled in 2018 that we previously reported, when using the total census population as the denominator rather than registered deaths [6]. Importantly, the patterns of inequities are similar in both studies, with young adults, Māori, Pacific peoples and those living in the most deprived areas least likely to be enrolled, and therefore least likely to access affordable PHC, and in results from the New Zealand Health Surveys these groups most frequently report financial barriers to care [27].

For babies, it is likely that the higher risk of mortality among those not enrolled reflects higher mortality among babies who are under hospital care, and do not have a chance to be enrolled. For young people, not being enrolled is protective when compared to those who are enrolled (see Table 4). However, being enrolled shows a demonstrated health need and it may be that the enrolled group have lower health status, on average. It is likely that unenrolled young people would acquire health benefits from enrolling. There is likely "adverse selection" going on. Ill young people do enrol to get health care but apparently-healthy young people don't as they perceive they don't need health care. Young people would benefit from enrolling, even if well, because of testing for unobserved diseases, reduced price contraceptives, direction to secondary and maternal services etc.

For older people, being enrolled is clearly protective and this could in part be due to the main cause of amenable deaths in the 45+ age groups being from cardio-vascular diseases (see S2 Appendix). Furthermore, some of the conditions included in the definition of amenable mortality could relate more to socio-economic factors than healthcare [28].

## Conclusions

The results of this study implicate a renewed focus on reaching those people who are not enrolled with a PHC provider. Given demonstrated inequities in enrolment levels across

socio-demographic groups, and in particular for Māori, efforts to improve enrolment could have significant benefits on health equity. As the health system reforms continue to be rolled out in Aotearoa New Zealand, the emphasis on ensuring a PHC system that is accessible to all must remain at the forefront of systems design.

## Supporting information

**S1 Appendix. List of amenable conditions, Ministry of Health 2018.**
(DOCX)

**S2 Appendix. Socio-demographic characteristics of amenable mortality cases by cause of death.**
(DOCX)

**S3 Appendix. Socio-demographic characteristics of the study population by enrolment status at death (premature deaths), 2008–2017.**
(DOCX)

## Acknowledgments

We acknowledge the National Collections team at the Ministry of Health NZ, who provided the dataset required for this analysis.

## Author Contributions

**Conceptualization:** Pushkar Silwal, Maite Irurzun Lopez, Mona Jeffreys.

**Data curation:** Pushkar Silwal, Maite Irurzun Lopez, Megan Pledger, Mona Jeffreys.

**Formal analysis:** Pushkar Silwal.

**Methodology:** Pushkar Silwal, Maite Irurzun Lopez, Megan Pledger, Mona Jeffreys.

**Project administration:** Maite Irurzun Lopez.

**Resources:** Maite Irurzun Lopez.

**Software:** Pushkar Silwal.

**Supervision:** Maite Irurzun Lopez, Megan Pledger, Jacqueline Cumming, Mona Jeffreys.

**Validation:** Jacqueline Cumming.

**Writing – original draft:** Pushkar Silwal, Maite Irurzun Lopez, Megan Pledger, Mona Jeffreys.

**Writing – review & editing:** Pushkar Silwal, Jacqueline Cumming.

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
