## [Decision Letter · Decision Letter 0]

10 Oct 2022

PONE-D-22-19204Association between enrolment with a Primary Health Care provider and amenable mortality: A population-based analysis in Aotearoa New ZealandPLOS ONE

Dear Dr. Silwal,

Thank you for submitting your manuscript to PLOS ONE. After careful consideration, we feel that it has merit but does not fully meet PLOS ONE’s publication criteria as it currently stands. Therefore, we invite you to submit a revised version of the manuscript that addresses the points raised during the review process.

Please, respond to all the comments of the reviewers. The description of the characteristics of the population not enrolled is a relevant topic in the manuscript. It is presented as a purpose of the study in the abstract. Also, in the discussion, the authors report inequities in enrolment by ethnicity. However, in results section, there is not a clear description of the association of enrolment with ethnicity and other variables.

Besides, the association between enrolment and amenable mortality, stratified by age, needs further description. The more significant differences (OR=4.9) are observed under 1 year of age. As expected, the most common causes of death in such group are related to perinatal conditions (Appendix 2). It seems likely that most of such newborns died in hospitals without any contact with primary care, and the enrolment data correspond to their mothers. Please, clarify this point.

Also, ethnicity and socioeconomic groups diverge in the causes of amenable deaths. There are noticeable differences in “Maternal and Infant Deaths” (almost limited to the perinatal ones). In my view, these findings could be developed.

Moreover, observe the PLOS Data Policy. PLOS journals require authors to make all data necessary to replicate their study’s findings publicly available without restriction at the time of publication. When specific legal or ethical restrictions prohibit public sharing of a data set, authors must indicate how others may obtain access to the data.

We look forward to receiving your revised manuscript.

Kind regards,

Juan F. Orueta, MD, PhD

Academic Editor

PLOS ONE

Journal Requirements:

2.  Please confirm your data is publicly available or if not what approvals you gained to access the data

“This study was conducted as part of the five-year Health Research Council New Zealand funded primary health care programme (HRC 18/667) “Enhancing primary health care services to improve health in Aotearoa New Zealand”.”

Reviewers' comments:

Reviewer's Responses to Questions

**Comments to the Author**

1. Is the manuscript technically sound, and do the data support the conclusions?

Reviewer #1: Yes

Reviewer #2: No

2. Has the statistical analysis been performed appropriately and rigorously? 

Reviewer #1: Yes

Reviewer #2: No

3. Have the authors made all data underlying the findings in their manuscript fully available?

Reviewer #1: No

Reviewer #2: Yes

4. Is the manuscript presented in an intelligible fashion and written in standard English?

Reviewer #1: Yes

Reviewer #2: Yes

5. Review Comments to the Author

Reviewer #1: Really interesting paper, using available data and appropriate statistical techniques to answer important questions.

I do have some comments, but I hope these will improve the paper.

In the abstract you state: We examine the characteristics of populations not enrolled, and whether enrolment is associated with amenable mortality.

In the introduction you write: The purpose of this analysis is to quantify the impact of not being enrolled with a PHO in Aotearoa New Zealand on amenable mortality, i.e. mortality for conditions that are amenable to health care interventions

And I think the tables do answer these aims.

Table 1: Socio-demographic characteristics of the study population by premature and amenable death status

Table 2: Socio-demographic characteristics of the study population by enrolment status

Table 3: Socio-demographic characteristics of amenable mortality cases and ORs of amenable deaths versus nonamenable deaths

However, I am not sure whether we need the detail on premature mortality in the paper, if you are not going to model this. But this is OK. I am more interested in why you didn’t model the factors associated with enrolment. In the discussion you talk about the groups who are less likely to be enrolled, and I am really interested in whether there are interactions between age, sex and socio-economic status and ethnicity. It is OK that you didn’t do this, but I wonder whether it would make some of your discussion stronger.

Table 4 – I am not clear what you have done. Have you run separate models for each age group, or used an interaction term. I have understood the OR to be for ‘enrolled V not-unrolled’. I am concerned that the enrolled V not enrolled in the youngest age group being associated with mortality may be complicated by babies dying before they have a chance to be enrolled. Have you thought about this?

In the discussion you say:

This could explain in part the differential results that we see by ethnicity, where Māori are less likely to enrol with a PHC provider, and experience higher rates of amenable mortality.

I am not sure why you don’t include/separately discuss Pacific peoples in the section, who are also more likely to die prematurely, and their death classed as amenable.

In the discussion, I think you should highlight that the definition of amenable might be a limitation, it is a govt definition, so fine to use, but presumably not everyone would agree. I am sure some people might argue that some of the conditions are more affected by socio-economic factors and not healthcare.

Minor Comments

Methods – The sentence about merging European and Asian was difficult to understand, I think you merged the two groups because the Asian group was small.

Table 1 has a footnote 3 and I am not sure that I could find it in the table.

Table 2 – can you make foot note 4 column percent by group? Or something equivalent

Please look at the heading and subject matter of this paragraph – it doesn’t seem match.

Relationship between primary health care enrolment and premature mortality

The adjusted odds of amenable mortality among those not enroled are 38.5% higher [OR=1.38, 95% CI 1.30-1.47] than those enrolled (Table 3). Similarly, males [OR 1.1, 95% CI 1.07-1.12] compared to females, youths aged 15-24 years [OR 6.37, 95% CI 5.18-7.88] compared to those aged 01-04 years, Pacific People and Māori [OR 1.32, 95% CI 1.25-1.38 and 1.22, 1.19-1.26 respectively] compared to nonMāori, non-Pacific people, and those living in the most deprived areas [OR=1.12, 95% CI 1.06-1.19] compared to those in the least deprived areas are more likely to die from amenable causes.

Can you clarify whether only the under 75 deaths were included in the amenable V non-amenable models? I think you may have said, but couldn’t find.

Apologies, but I had never seen ‘Aotearoa New Zealand’ before. I therefore did not realise it was a national study. I am not sure of the best approach to this for a less informed audience. In the title could the word national be inserted before Aotearoa New Zealand?

Reviewer #2: 1. Causal language

“The purpose of this analysis is to quantify the impact of not being enrolled with a PHO in Aotearoa New Zealand on amenable mortality”

“This confirms our initial hypothesis that not being enrolled and, thus, missing out on the benefits

that enrolling provides, will impact negatively on health outcomes.”

I suggests authors refrain from using the language that implies causality in the association between PHC enrollment and amenable mortality.

2. Descriptive statistics tables in the manuscript itself and in the appendix to the manuscript depict row statistics. This makes it much harder to a reader to understand how characteristics of the enrolled and not enrolled subsamples differ. I suggest authors convert tables to column percentages everywhere not just for “total” column.

3. Stemming from point 2 above the section of the paper that should have described how demographic characteristics of the enrolled subsample differ from those of not enrolled subsample is murky. There is chi square test present in Table 2 but it’s unclear what this test actually compares to what. Manuscript needs a clear section that would describe how gender, age, ethnicity and area deprivation composition of the enrolled subsample differs from not enrolled.

4. Manuscript simply presents adjusted associations between PHC enrollment and amenable mortality. The authors never make a case why these estimates should be interpreted as causal while the language of the paper suggests it should. Authors should either show why it’s appropriate to interpret their estimates as causal or admit that the estimates are not causal.

5. Table 2 shows that association between not enrolled and amenable mortality is statistically significant but the sign of this association in the bivariate and adjusted models are opposite of one another. Why is that?

6. According to table 3, 15-25 year old adolescents and young adults are likely to die from amenable causes than 5-14 year old children. According to table 4, 15-24 year old adolescents and young adults who are not enrolled are less likely to die from amenable causes. Could authors discuss this result? Since authors interpret results as causal would they suggest that disenrolling youth starting age 15 from primary health care provider would save their lives?

6. PLOS authors have the option to publish the peer review history of their article (what does this mean?). If published, this will include your full peer review and any attached files.

Reviewer #1: **Yes: **Kate Honeyford

Reviewer #2: No

---

## [Author Response · Author response to Decision Letter 0]

27 Dec 2022

Thank you so much for your time reviewing our manuscript. We appreciate your comments and suggestions. We have addressed the comments to the extent possible and/or responded to them as outlined in ‘Response to Reviewer’ document uploaded into the portal.

---

## [Decision Letter · Decision Letter 1]

17 Jan 2023

Association between enrollment with a Primary Health Care provider and amenable mortality: A national population-based analysis in Aotearoa New Zealand.

PONE-D-22-19204R1

Dear Dr. Silwal,

We’re pleased to inform you that your manuscript has been judged scientifically suitable for publication and will be formally accepted for publication once it meets all outstanding technical requirements.

Kind regards,

Juan F. Orueta, MD, PhD

Academic Editor

PLOS ONE

Additional Editor Comments

I have noticed a relevant typo in the results section (Lines 219-222). The text describes the mortality percentages (overall and premature) among NOT enrolled individuals (instead of the enrolled ones).

Reviewers' comments:

Reviewer's Responses to Questions

**Comments to the Author**

1. If the authors have adequately addressed your comments raised in a previous round of review and you feel that this manuscript is now acceptable for publication, you may indicate that here to bypass the “Comments to the Author” section, enter your conflict of interest statement in the “Confidential to Editor” section, and submit your "Accept" recommendation.

Reviewer #1: All comments have been addressed

2. Is the manuscript technically sound, and do the data support the conclusions?

Reviewer #1: Yes

3. Has the statistical analysis been performed appropriately and rigorously? 

Reviewer #1: Yes

4. Have the authors made all data underlying the findings in their manuscript fully available?

Reviewer #1: No

5. Is the manuscript presented in an intelligible fashion and written in standard English?

Reviewer #1: Yes

6. Review Comments to the Author

Reviewer #1: Thank you for responding to my comments. Please check that the spelling of enrolled is consistent throughout the paper.

7. PLOS authors have the option to publish the peer review history of their article (what does this mean?). If published, this will include your full peer review and any attached files.

Reviewer #1: No

---

## [Editor Report · Acceptance letter]

25 Jan 2023

PONE-D-22-19204R1 

Association between enrolment with a Primary Health Care provider and amenable mortality: A national population-based analysis in Aotearoa New Zealand 

Dear Dr. Silwal:

I'm pleased to inform you that your manuscript has been deemed suitable for publication in PLOS ONE. Congratulations! Your manuscript is now with our production department. 

Kind regards, 

on behalf of

Dr. Juan F. Orueta 

Academic Editor

PLOS ONE